# Histamine Control in Raw and Processed Tuna: A Rapid Tool Based on NIR Spectroscopy

**DOI:** 10.3390/foods10040885

**Published:** 2021-04-18

**Authors:** Sergio Ghidini, Luca Maria Chiesa, Sara Panseri, Maria Olga Varrà, Adriana Ianieri, Davide Pessina, Emanuela Zanardi

**Affiliations:** 1Department of Food and Drug, University of Parma, Strada del Taglio 10, 43126 Parma, Italy; sergio.ghidini@unipr.it (S.G.); mariaolga.varra@unipr.it (M.O.V.); adriana.ianieri@unipr.it (A.I.); emanuela.zanardi@unipr.it (E.Z.); 2Department of Health, Animal Science and Food Safety, University of Milan, 20133 Milan, Italy; luca.chiesa@unimi.it; 3Quality Department, Italian Retail Il Gigante SpA, 20133 Milan, Italy; pessina@ilgigante.net

**Keywords:** food contaminants, food safety, biogenic amines, rapid methods, chemometrics, fish control

## Abstract

The present study was designed to investigate whether near infrared (NIR) spectroscopy with minimal sample processing could be a suitable technique to rapidly measure histamine levels in raw and processed tuna fish. Calibration models based on orthogonal partial least square regression (OPLSR) were built to predict histamine in the range 10–1000 mg kg^−1^ using the 1000–2500 nm NIR spectra of artificially-contaminated fish. The two models were then validated using a new set of naturally contaminated samples in which histamine content was determined by conventional high-performance liquid chromatography (HPLC) analysis. As for calibration results, coefficient of determination (*r*^2^) > 0.98, root mean square of estimation (RMSEE) ≤ 5 mg kg^−1^ and root mean square of cross-validation (RMSECV) ≤ 6 mg kg^−1^ were achieved. Both models were optimal also in the validation stage, showing *r*^2^ values > 0.97, root mean square errors of prediction (RMSEP) ≤ 10 mg kg^−1^ and relative range error (RER) ≥ 25, with better results showed by the model for processed fish. The promising results achieved suggest NIR spectroscopy as an implemental analytical solution in fish industries and markets to effectively determine histamine amounts.

## 1. Introduction

The fishery and aquaculture sector has seen notable growth and development in recent years. This great expansion is partially justified by the increased demand for fish supply being, in turn, driven by the improvement of the logistics of goods transport, storage, free-trade agreements among countries and the growing general awareness of beneficial effects deriving from fish consumption.

The global production of tuna and tuna-like fish species is estimated to account for 6 million tons per year [1] and its related international trading ranges from the marketing of raw material for direct consumption or further processing to the preserved canned product [2]. In common with other fishery products, tuna is particularly prone to fast deterioration and chemical and microbiological contaminations [2,3], which together increase the likelihood of health risks associated with its consumption. Poisoning deriving from the consumption of spoiled or bacterially contaminated fish with high amount of histamine is the most common foodborne illness associated with fish consumption in the world and it is of particular concern due to the severe toxic effects generally occurring at histamine levels of more than 200 mg kg^−1^ [4,5]. For this reason, histamine is the only biogenic amine for which regulatory levels have been established at the international level [6,7]. Histamine formation in fish products is mainly driven by the presence of histidine in the product and the activity of the microbial histidine decarboxylase enzyme. At the same time, any factor interacting with the formation and the activity of the enzyme is responsible for the increase of histamine concentrations [8]

The nature and composition of the raw material (i.e., pH, ion strength, and high amounts of free histidine into flesh of Clupeidae, Scombridae, Scomberesocidae, Pomatomidae and Coryphaenidae families), microorganisms (in terms of both microbial load and type of microorganisms) and handling and storage conditions (preservation temperatures, canning, fermentation, smoking, etc.) are most influential aspects affecting the formation of histamine in fish muscle [8,9]. Once produced in raw fish, histamine is almost irremovable due to its thermostability and, thus, common heating used for fish cooking or processing are ineffective for its removal [8,10].

A further problematic aspect of the presence of histamine in fish and seafood is related to the fact that histamine has not any correlation with sensory changes of fish products [8]. Therefore, histamine is not perceivable by human senses and this may worsen the hazard of ingestion exposure of consumers [10].

Therefore, good hygienic practices and temperature control from fish catch to manipulation, storage and commercialization represent the main preventive measures to be adopted to reduce the risk associated with histamine, followed by the enforcement of action levels for histamine detection by official surveillance programs and inspections [9]. In particular, the control of histamine levels of tuna is highly relevant in the context of fraud related to the use of carbon monoxide to retain the red color of the fish beyond its microbiological shelf life. In the period between 2010 and 2020, 73 notifications were recorded in the Rapid Alert System for Food and Feed (RASFF) portal for carbon monoxide treatment or suspicion of treatment in tuna [11].

Historically, several analytical methods have been developed for monitoring histamine concentrations in food items, which are mainly based on electrophoretic [12], fluorometric [13], enzymatic or immunoassay detection [14] and chromatographic separation by thin-layer chromatography [15], high performance liquid chromatography (HPLC) [16] or gas chromatography [17], and, recently, also biosensors [18].

Among these techniques the most used one is the AOAC 977.13 fluorometric method established by the Codex Alimentarius [19], although the official reference method in Europe for histamine testing in fish and seafood mandated by the Regulation (EC) no. 2073/2005 [6] is based on HPLC coupled to UV detection [20]. Despite many of these being accurate, specific, precise and well consolidate methods, they still suffer from high price of the instruments, facilities and/or regents, consumption of large amounts of both solvents and samples, sample destructiveness, preliminary sample preparation steps and/or elaborate post-treatments of the analyte, longer time for analysis, and the demand for operator expertise [20]. Therefore, what is currently most demanded is the development of faster, easier, cheaper and ecofriendly methods able to meet the needs of modern industry [21,22].

In such a scenario, vibrational spectroscopy, when coupled with innovative chemometrics or machine learning techniques, can be regarded as forthcoming essential tools for fish and seafood monitoring. Near infrared (NIR) spectroscopy has been recently used as a qualitative method to discover fraud and ascertain the integrity of different foods of animal origin. The identification of the adulteration of lamb, beef [23] and milk [24], the use of unauthorized preservation methods in fermented meat products [25], detection of spoilage bacteria in pork meat [26] and mislabeling of eggs in relation to production process [27] and honey [28] in relation to the geographical origin are just some examples. As for specific applications to fish and seafood, NIR spectroscopy has been successfully applied to qualitatively discriminate many products according to different authenticity features [29,30,31,32,33,34,35,36], but also to quantify major, minor and trace matrix components [37,38,39,40,41,42,43,44,45,46,47,48,49,50,51]. A short overview of the principal applications of NIR spectroscopy to the fish and seafood sector is provided in Appendix A. Specifically, minor and trace components quantification by NIR spectroscopy is usually complicated by the low sensitivity of the NIR instrumentation and the interferences caused by the water content of food. Nevertheless, it has been clearly demonstrated that proper experimental design and spectral processing techniques can led to optimal NIR-based calibration for the quantification of minor and trace components of fish at the ppm- or ppb-level, such as total volatile basic nitrogen (TVB-N) [37], malondialdehyde [38] and pigments [39].

Some applications of vibrational spectroscopy aimed to quantify histamine in fish matrices are also available. Surface-enhanced Raman spectroscopy (SERS) [52,53,54] and dry extract system for infrared (DESIR) coupled with NIR spectroscopy [39] demonstrated a great potential in overcoming the low sensitivity typically associated with these techniques, by amplifying the histamine signal through plasmonic resonance and preconcentrations steps, respectively. Although the very promising results achieved, SERS is characterized by high cost of the equipment, while DESIR suffers from long time for analysis.

Based on the need for tightening up fish controls and shifting towards a risk-based control of fish safety, the implementation of up-to-date, rapid, and efficient food inspection tools may represent an important contribution to support official methods. Hence, the aim of the present research was to evaluate for the first time the feasibility of NIR to directly quantify histamine content in raw and processed tuna according to maximum levels for fish, with minimal processing of fish samples. This could represent a great advantage in terms of reduced workload and saving of samples for both fish industries and fish markets, in which rapid information toward food safety play a pivotal role in improving food management and avoiding food losses, and for competent authorities to increase the number of checks.

## 2. Materials and Methods

### 2.1. Chemicals

Perchloric acid, ammonium acetate, acetone, ammonia (25%) and acetonitrile were purchased from Merck (Darmstadt, Germany). The internal standard (IS, 1,3-diaminopropane) was purchased from Sigma (St Louis, MO, USA), histamine dihydrochloride (99%) and dansyl chloride—used as a labeling reagent for the derivatization—were provided by Acros Organics (Geel, Belgium). All the reagents employed for HPLC analysis were HPLC-grade. Deionized water was purified by a Milli-Q water system (Millipore Corp., Bedford, MA, USA). Primary stock solution of 10 g L^−1^ of histamine was prepared by appropriate dissolution of histamine dihydrochloride standard in deionized water. Spiking standard solutions used for calibration were prepared by appropriate weekly dilution of the primary stock solution to obtain final known concentrations of 10, 50, 100, 200, 400 and 1000 mg L^−1^ of pure histamine in a spiking volume of 0.1 mL.

### 2.2. Sample Collection and Experimental Design

NIR calibration models were created by collecting 12 fillets of raw tuna fish belonging to the species *Thunnus albacares* from five different origins (FAO fishing areas 87, 77, 34, 51 and 31), and 12 samples of processed tuna fish (canned in olive oil or salt water) obtained through the processing of the species *Katsuwonus pelamis* or *Thunnus albacares* from three different areas (FAO 71, 31 and 34). Each sample was minced and homogenized by using a domestic blender, after draining off the oil from canned. After that, 4 raw tuna pools and 4 processed tuna pools were created by randomly mixing 3 minced samples to obtain final calibration working samples (calibration samples).

Overall, 40 raw tuna fillets and 40 processed samples (canned in olive oil or salt water) provided from the Milan (Italy) fish market and food industry were included into the study and used as real samples to validate the calibration models (validation samples). Each validation sample was minced and homogenized by using a domestic blender (oil or salt water from canned muscle was firstly drained off), and the histamine content predicted by NIR spectroscopy. All validation samples were processed also by using the official reference HPLC method to confirm the feasibility use of the NIR technique as a rapid tool to assess fish safety criteria. In order to check for robustness, stability and reproducibility of the NIR-based histamine quantification method over samples’ shelf life, 10 samples of raw tuna were randomly chosen, stored at room temperature (20 ± 1 °C) for 3 days and subsequently remeasured by HPLC and NIR spectroscopy.

A graphical flowchart summarizing the main steps of the experimental procedure applied is provided in Figure 1.

### 2.3. Reference HPLC Analysis

Histamine content was determined according to Duflos et al. [55] and Chiesa et al. [56]. Briefly, 4 g of raw or processed tuna samples were weighed and, after adding 250 µL of 1 mM IS and 10 mL of 0.4 M perchloric acid solution, homogenized and centrifuged for 10 min at 2400× *g*. The supernatant was transferred into a 25 mL bottle through filter paper. The extraction was repeated with 10 mL of 0.4 M perchloric acid solution and centrifuged. The two supernatant aliquots were merged. Derivatization procedure of biogenic amines with dansyl chloride solution (1 mL, 10 mg mL^−1^ in acetone) was performed at 40 °C for 45 min. Excess of the derivatization reagent was finally neutralized by adding 100 µL ammonia (25%). After 30 min, final extract was adjusted to 5 mL with 0.1 M of ammonium acetate/acetonitrile (1:1) and filtered using a 0.45 µm syringe filter (Sartorius, Goettingen, Germany).

The histamine content was determined using a HPLC Jasco quaternary pump (Ishikawa-cho, Japan) equipped with the autosampler. An adequate elution gradient of ammonium acetate (0.1 M) and acetonitrile was applied as mobile phase with a flow rate of 1 mL min^−1^. Spherisorb ODS-2, 5 µm, 125 mm × 4 mm (Waters Corporation, Milford, MA, USA) reverse-phase column was used while detection was carried by UV/VIS detector (Jasco, Ishikawa-cho, Japan) operating at 254 nm.

### 2.4. NIR Apparatus and Method

After having assessed the absence of histamine and other biogenic amines in each tuna pool by HPLC (histamine < limit of quantification, LOQ), samples intended for NIR calibration were aliquoted into 1 gram-subsamples and placed into a 32 mm diameter round-bottom optical glass cuvette. Histamine solutions (0.1 mL) at the range of concentrations of 0, 10, 50, 100, 200, 400 and 1000 mg L^−1^ of histamine were further introduced into the cuvette (Figure 1). Spiked subsamples were manually mixed with a spoon and were let to absorb the solution for 2 min. Diffuse reflectance NIR spectra were recorded by using a NIRFlex^®^ N-500 FT-NIR spectrometer (Büchi Labortechnik AG, Flawil, Switzerland). A total of 140 calibration subsamples were scanned (both for raw and processed samples), corresponding to 5 replicates for each of the 7 histamine-spiking levels for each of the four tuna fish pools created. Each spectrum was recorded at a 1 nm interval for wavelengths between 1000 and 2500 nm as the average of 64 coadded scans. Four different spectra were acquired for each calibration sample by changing the position of the cuvette into the spectral windows of the instrument to include possible inhomogeneities deriving from sample preparation. Absorbance values for each wavelength were finally obtained as the logarithm of the inverse of reflectance values recorded and exported for the setup of the calibration quantitative models.

### 2.5. NIR Data Processing

The absence of spectral outliers was proven through the application of Hotelling T² statistics (95% confidence interval). After that, the four spectral replicates for each sample were averaged to get a final representative spectrum.

Calibration equations for the quantification of histamine in tuna samples were developed by using SIMCA 16.0.2 software (Sartorius Stedim Data Analytics AB, Umea, Sweden). Orthogonal partial least square regression (OPLSR) was hence selected as algorithm to correlate the whole 1000–2500 nm spectra (independent variables, X-matrix) and the histamine spiking levels (dependent variable, Y-matrix).

At first instance, different combinations of chained spectral filters were investigated to correct light scatter effects, baseline offset, linear trend and noise in the raw spectra. After having tested different algorithm combinations, standard normal variate (SNV) plus second derivative (2D) using the Savitzky–Golay algorithm (SG) (quadratic polynomial fit to fifteen points) was chosen as the best filtering combination since the lowest calibration-associated errors and the best fitting and prediction outcomes (described below) were obtained.

The corrected X-matrix and the Y-matrix were subsequently mean-centered. A 7-fold cross validation (CV) was chosen to ensure the stability of the calibration model and to select the optimal number of predictive and orthogonal partial least square (OPLS) factors to retain. The root-mean square error of cross-validation (RMSECV), root-mean square error of estimation (RMSEE), the coefficients of determination R^2^X and R^2^Y (representing the fraction of variance of the X-matrix and Y-matrix explained by the regression model), and the prediction ability summed up by the Q^2^ value were evaluated to estimate the quality of the calibration model and to select the most performant one.

#### Construction and Validation of NIR Models for Histamine Quantification

The quality of the final OPLSR calibration models for raw and processed tuna samples was evaluated by analyzing the analytical figures of merit reported below.

The linearity of the calibration equations was evaluated taking into consideration the coefficient of determination (*r*^2^), the slope and the intercept of the regression line.

Intraday precision (repeatability) and interday precision (reproducibility) were estimated by acquiring the NIR spectra of the same quality control (QC) sample spiked with 50 mg kg^−1^ of histamine six times consecutively in the same day and six times throughout three different days, respectively. During the 3 day-storage period, samples were kept at a temperature close to 0 °C. The results were reported as relative standard deviation (RSD, %) of the 6 and the 18 repeated measurements.

The limit of detection (LOD) and limit of quantification (LOQ) parameters, used to evaluate the sensitivity of the regression models, were respectively calculated as three and ten times the standard deviation of repeated blank measurements divided the slope value of the calibration regression equation.

The evaluation of the specificity was carried out through the analysis of the degree of concordance between the amplitude of the regression coefficients and the characteristic absorbance bands of the pure histamine NIR spectrum. The accuracy of the calibration models, expressed as recovery values (%), was assessed by comparing the histamine content predicted by the NIR model and the actual reference values of five different QC samples spiked at the specific target levels of 10, 50 and 100 mg kg^−1^ and quantified via HPLC.

Supportive evidence of specificity and accuracy of the two final OPLSR calibration models for histamine quantification was also provided through the prediction of histamine content in validation samples listed in Section 2.4. Briefly, after homogenization, 1 g aliquots of raw or processed tuna fish samples were analyzed by NIR spectroscopy, following the analytical procedures previously described (Section 2.4) and briefly summarized in Figure 1. Each validation sample was analyzed in duplicate, and four spectra were acquired for each replicate. The NIR prediction of the histamine concentration was performed using the equations developed during the calibration stage, and the quantitative results were compared with those obtained by the official reference HPLC method.

The root-mean square error of prediction (RMSEP) was used to evaluate the accuracy in prediction of the NIR models. The range error ratio (RER) calculated as the ratio between the ranges of NIR predicted values in the validation set and the RMSEP [57] was also employed to this purpose, considering the intervals proposed by Williams [58]. Specifically, RER values lower than 7 indicate poor models, between 7 and 12 good models for approximate screening, between 13 and 20 good models for screening and higher than 21 models ideal for quality control.

Raw tuna samples stored at room temperature (20 ± 1 °C) for 3 days and intended for models’ validation over possible modification during shelf-life were equally processed and analyzed. The results were compared with those obtained at the time of the first analysis.

## 3. Results and Discussion

### 3.1. Histamine Quantitative Analysis by HPLC

Histamine quantification by HPLC was performed on validation samples to verify wheatear NIR predictions were accurate enough compared to reference the HPLC method.

The HPLC method led to linear calibration lines with *r*^2^ value of 0.9936 in the 5–1500 mg kg^−1^ histamine range. The limit of detection (LOD) and the limit of quantification (LOQ) for histamine were 0.003 and 0.01 mg kg^−1^ histamine, respectively.

### 3.2. Performances of the NIR Calibration Methods for the Prediction of Histamine

Histamine is the only biogenic amine for which regulatory maximum limits of 200 mg kg^−1^ and 400 mg kg^−1^ in fresh and enzyme matured fish, respectively, have been set by the European Commission Regulation (EC) No 2073/2005 [6] and its amendment Regulation (EC) No 1441/2008 [59]. Tuna fish, in particular, has been reported as the fishery product most frequently non-compliant for histamine, according to the notifications of the RASFF [60,61]. Although these legal requirements on the maximum limits, alert notifications have recently revealed histamine concentrations in fish and seafood products higher than 2000 mg kg^−1^ [60,61,62]. Hence, the calibration model for histamine in tuna samples developed in this study was built to predict the concentration of histamine up to 1000 mg kg^−1^. The goodness of the developed calibration models is summarized in Table 1. Briefly, 1 predictive and 4 or 5 orthogonal OPLS factors were calculated for raw and processed tuna models, respectively. In both the situations, the cumulative values of R^2^X, R^2^Y and Q^2^ were considered suitable to define the model robust enough. In particular, the predictability values (Q^2^) close to 1 imply the great potential of the models in properly quantifying histamine content. Likewise, the similar magnitude of RMSECV and RMSEE values obtained excludes potential troubles, which could have resulted from the cross-validation procedure.

The calibration equations for histamine quantification obtained by the application of the OPLSR to the NIR spectra of raw and processed tuna samples are reported below (Equations (1) and (2)):OPLSR model for raw samples: Y = x − 2.942 × 10^−7^(1)
OPLSR model for processed samples: Y = x + 1.768 × 10^−5^(2)

Samples distributing linearly around the regression lines reported in the predicted vs. observed values graph (Figure 2A,C), the *r*^2^ values for raw (*r*^2^ = 0.987) and processed tuna (*r*^2^ = 0.987) regression models, and, finally, the slope values equal to 1 (Table 1), suggest a perfect linearity in the range 0–1000 mg kg^−1^ and an excellent quantitative prediction [58]. Moreover, the Y-residuals plotted in Figure 2B and Figure 2D were normally distributed, and none of the data point fell outside the three standard deviation boundaries, thus confirming an acceptable data fit.

When using NIR spectroscopy to predict chemical properties and quantify components in foodstuffs, an *r*^2^ value higher than 0.95 is considered an excellent metric of the quality of the model [63]. Nonetheless, the likelihood of achieving such optimal results increases with the relative amount of the component to be quantified, so much so that *r*^2^ values higher than 0.95 are frequently attained in those applications dealing with the determination of proximate composition of fish [41,42]. Although histamine in fish is present at the ppm level, the reliability of obtaining *r*^2^ > 0.90 may be confidently associated to the appropriateness of the preprocessing techniques, which are reported to improve the linear relationship between the spectral signals and analyte concentrations [64] and has been confirmed for different food contaminants present at the same concentration range or lower than those of biogenic amines. For example, the cadaverine contents in mackerel muscle [51], or residues of Penicillin G in milk samples [65] were predicted by developing NIR calibration straight lines with *r*^2^ values being higher than 0.98. Similarly, *r*^2^ value > 0.97 were obtained for the quantification of deoxynivalenol [66] and aflatoxin B1 [67] in foods of plant origin.

The variability between results from repeated measurements on the same sample during the same day or during subsequent days also demonstrated a good precision of the calibration models. The best repeatability and reproducibility outcomes were obtained for the model predicting histamine in processed tuna samples, corresponding to RSD values of 3.2% and 4.8%, respectively (Table 1). Even in the case of the raw tuna-model, reproducibility values (8.1%) were worse than repeatability values (4.9%) due to the changes in tuna composition that despite refrigeration, occur during the storage of samples. Similarly, mean percentages of recovery ranging from 98.2% and 108.4% were calculated for NIR-based model created for processed tuna samples (see Table 1), thus indicating a satisfying degree of accuracy for the method.

The LOD values for histamine were 4 mg kg^−1^ and 2 mg kg^−1^ in raw and processed tuna fish samples, while the LOQ values were calculated at 12 mg kg^−1^ and 8 mg kg^−1^, respectively. Other authors reported significantly better results when developing NIR predictive models for the quantification of trimethylamine (TMA-N) in fish, by achieving LOD and LOQ values of 0.014 mg (N) kg^−1^ and 0.021 mg (N) kg^−1^ [45]. Although the results of this study did not suggest such an optimal sensitivity of the quantification method, they were considered suitable for use, especially when it comes to controlling histamine at the range of concern (above 100 mg kg^−1^). Hence, sensitivity results achieved in the present work may be assumed to be the consequence the preprocessing techniques applied to the spectra [68] and of the indirect relation among the overall NIR spectral fingerprint and histamine interacting and covarying with other major or minor components of fish.

The specificity of the two regression models was graphically investigated by analyzing the influence of each NIR wavelength on Y-matrix by the regression coefficients, so as to exclude possible accidental correlations [69].

A satisfying degree of concordance between the characteristic NIR absorption bands of pure histamine and the highest regression coefficients was observed for the two quantitative models (Figure 3).

In particular, the highest contribution was exerted by the regression coefficients of the NIR wavelengths beyond the 2200 nm region (Figure 3B,C), which corresponded to the intense absorption bands of histamine in the 2240–2450 nm region (Figure 3A). In this region, the peak around 2475 nm can be directly attributed to symmetric stretching vibrations of the C-N-C bonds typical of nitrogen containing compounds as histamine [70] whose chemical structure is illustrated in Figure 3A. Besides, the peak at 2240 nm has been already related to N-H stretching vibrations of amino acids and recently identified as a characteristic feature of the NIR spectrum of histamine [40]. High regression coefficients were also observed for raw tuna fish-based model in the 2020–2200 nm region, where the peak at 2180 is the consequence of bending vibration of the N–H bond during the second overtone [70]. On the contrary, in the processed tuna fish-based model higher positive and negative regression coefficients appeared around 1280–1400 nm and 1600–1700 nm. Wavelengths falling within these ranges have been assigned to C-H and O-H vibrations typical of fats or aromatic compounds [70], and which could have resulted from the residual packing oils. Absorbances in the region 1666–1562 nm have been attributed to stretching vibrations of C-H, N-H and C≡N bonds [45]. Similarly, other bands around the 1930–2020 nm NIR region of processed tuna (e.g., 1936, 1951, 1979 and 2011) closely characterize fish minor and major compounds, having a link with stretching and combination vibrations of C-H, N-H and C≡C groups and with stretching vibrations of the C=O group [33,42,66].

The overall calibration results discussed above might open a wide spectrum of possible applications of the methodology to fish and seafood safety control. Nevertheless, the performances of the multivariate methods rely on the way the calibration curve is constructed, i.e., the total number and the composition of the samples, and the concentration range of the analyte of interest included into the model. In this work, many sources of natural variability were introduced by considering samples of different origins and/or species, so as to make the models robust towards real scenario applications. However, experimentally contaminated samples rather than naturally contaminated ones were employed in the calibration stage, due to the difficulties in collecting the latter at high histamine concentration ranges. To overcome this aspect and verify whether the trained models are applicable to naturally contaminated fish samples, histamine concentrations of real market tuna samples were quantified by HPLC and concomitantly predicted by using the developed NIR calibration curves.

### 3.3. Comparison Between NIR and HPLC Analyses of Validation Samples

Histamine concentrations were determined by means of NIR spectroscopy and HPLC in 40 raw and 40 processed tuna fish samples retrieved from the market (listed in Section 2.2) to validate the NIR quantitative methods previously developed. The results obtained by using the two instrument systems are shown in Table 2. As it can be observed, a good degree of concordance between the HPLC and NIR quantifications was found, with Pearson’s correlation coefficients between the two methods of 0.615 and 0.689 for raw and processed validation tuna samples, respectively. Mean absolute errors of the NIR method ranging from 7 to 415 mg kg^−1^ were also achieved when predicting histamine in raw tuna. In this case, the lowest absolute error was observed for the samples characterized by the lowest histamine amounts detectable by both NIR and HPLC instruments, while the highest absolute error was calculated for the samples containing the highest amounts, predicted as 1694 ± 300 mg kg^−1^ (Table 2). Histamine amounts in processed tuna were instead predicted with a narrower range of absolute errors, varying between 8 and 30 mg kg^−1^, but NIR responses to concentrations in excess of 100 mg kg^−1^ were not assessable due to the lack of processed samples with concentrations higher than that. In connection to this, *r*^2^ values in prediction of 0.921 and 0.945 for raw and processed tuna were achieved, which despite slightly lower compared to the *r*^2^ metrics reported for calibrations, were generally considered satisfying.

Raw and processed validation samples also provided RMSEP values of 10 and 4 mg kg^−1^ which were in the same order of magnitude than RMECV and RMSEE achieved in calibration (see Table 1). Based on this and on similar information retrieved from the calculated RER values—which were 175 and 25 for raw and processed tuna, respectively—the great potential of the two NIR predictive models for screening and quality control applications was confirmed.

The ability of NIR spectroscopy to predict histamine content regardless of its freshness storage was also demonstrated by storing a total of 10 raw tuna samples at room temperature (20 ± 1 °C) for three days (Figure 4).

For most of the samples, histamine concentrations were only slightly affected by the storage time. This could be explained by the fact that the development of histamine in fish depends not only on the species and storage conditions, but also on the microbial type and load [71,72]. High histamine-producing bacterial strains are in fact more likely associated to the natural aquatic environments (and commonly found in gut, gills and skin of the fish) rather than to the manufacturing and storage environments [71,72]. Nevertheless, two samples were characterized by an increase of histamine up to more than 120 and 590 mg kg^−1^, as quantified by HPLC (Figure 4). In this case, also NIR response was satisfying as the concentration of 157 ± 30 and 734 ± 50 mg kg^−1^ were respectively predicted. These findings suggest the robustness and the specificity of the methodology also when modifications of the fish composition occur during the storage, thus hinting at the possibility of using the models for predicting histamine also in fish products at different stages of freshness during their shelf-life.

One of the most predominant peculiarity emerging by analyzing data is the tendency of NIR spectroscopy to overestimate histamine concentrations compared to the reference HPLC method. This NIR characteristic was previously observed for the prediction of other food constituents [73], and already attributed to possible interferences deriving with other minor components of complex matrices responsible for signal amplification [74]. Nevertheless, the magnification of histamine concentrations, which proportionally increased with increasing analyte amounts, might be also associated to the higher degree of inhomogeneity in the validation tuna samples compared to the pooled tuna samples created under laboratory conditions and used during the calibration stage.

Although overestimation of the histamine content might be to some extent problematic for the fish industry due to a potential increase in the number of non-compliant samples requiring confirmation by the official method, the experimentally achieved results suggest, however, the feasibility of the application of the method to assist fish safety and quality management systems. The proposed methodology would be indeed beneficial to fish industries and markets thanks to the improvement of quality and safety standards.

## 4. Conclusions

The results achieved in the present research work represent a proof of concept for the use of NIR spectroscopy as a reliable technique to predict the content of histamine in raw and processed tuna fish, opening up new opportunities for monitoring hazardous foodstuffs.

The proposed method, which is based on the use of the full 1000–2500 nm NIR spectra of fish and OPLS regression to correlate the spectral data with the reference values obtained by the reference HPLC method, can handle histamine contamination in a wide dynamic range at concentrations up to 1000 mg kg^−1^, and takes advantage from the minimal processing of solid samples, making no use of environmentally harmful solvents and time-consuming extraction or preconcentration steps.

At this stage of the method development, the observed slight overestimation of the histamine content by NIR spectroscopy may not represent a problem in the framework of fish inspection practices, since it would allow the identification of samples whose histamine content may be high. Whereas the non-destructiveness, rapidity, and high throughput of the technique result in an overall increase of the number of samples measured, the probability of non-compliant samples to be identified, collected and confirmed by the reference method also increases, thus leading to more targeted and optimized sampling plans for the overall improvement of food safety. For all these reasons, the method was proposed to be used in the short term as a sorting, screening and semiquantitative tool in support of the reference methods for histamine quantification, pending the robustness of the models and measurement uncertainties are improved over time by including further naturally incurred calibration and validation samples in the range of interest.

Looking forward, the possibility of extending the method also to the simultaneous quantification of other biogenic amines and transferring it to an automatable or portable NIR spectrometer would make the technique tailored for performing quantification directly “on-site” in fish processing and manufacturing plants, markets, catering, and food service industries. This would reduce time for analysis, sample consumption and economic losses, thus a having positive impact for fish business operators, stakeholders, public authorities and technical bodies, and for consumers.

## Figures and Tables

**Figure 1 foods-10-00885-f001:**
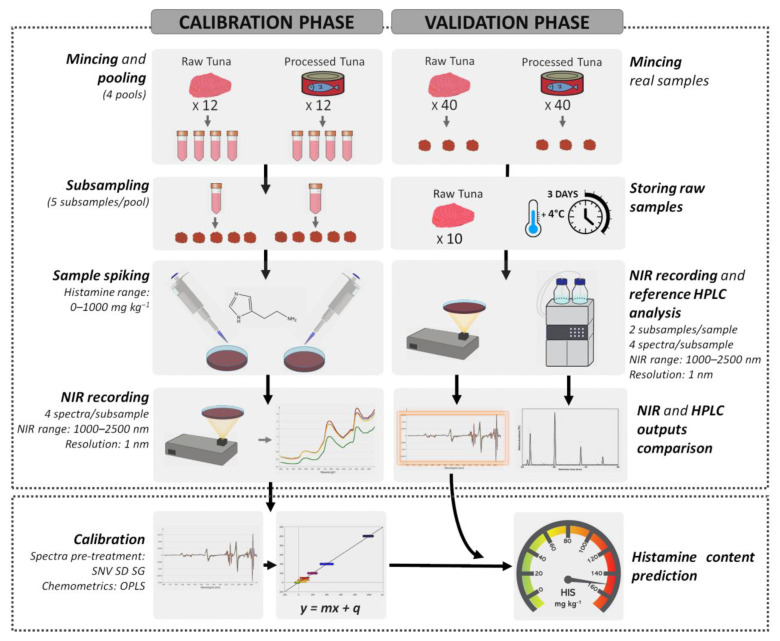
Flowchart summarizing the principal experimental procedure steps for the development of NIR spectroscopy models predicting histamine concentrations in raw and processed tuna samples.

**Figure 2 foods-10-00885-f002:**
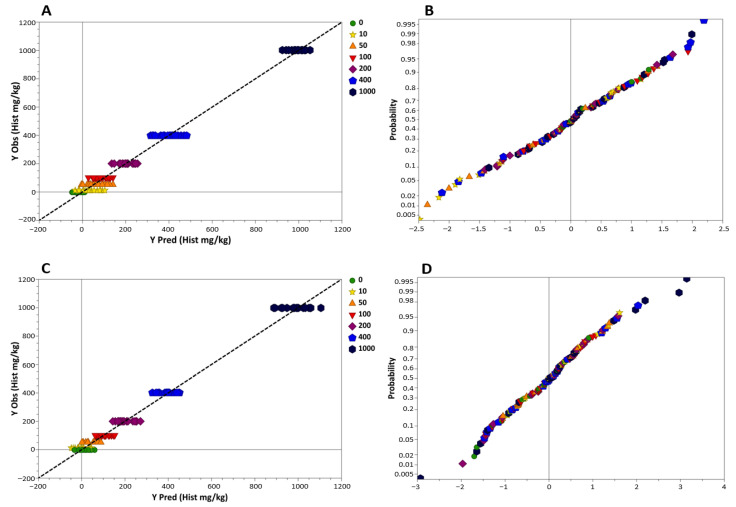
NIR-predicted vs. observed (actual) histamine values in the calibration regression model for raw tuna fish (**A**) and processed tuna fish (**C**) and relative residual normal probability plot (**B**,**D**). Histamine concentrations: green circles, blanks (mg kg^−1^); yellow stars, 10 mg kg^−1^; orange triangles, 50 mg kg^−1^; inverted red triangles, 100 mg kg^−1^; purple diamonds, 200 mg kg^−1^; blue pentagons, 400 mg kg^−1^; black hexagons, 1000 mg kg^−1^.

**Figure 3 foods-10-00885-f003:**
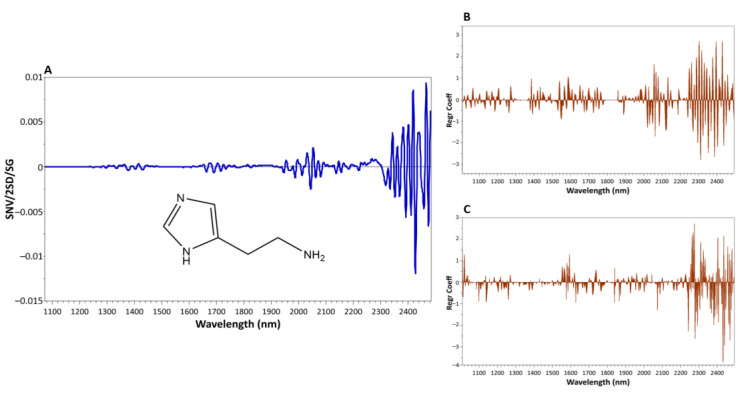
Mean processed spectrum (SNV + 2D + SG) plus chemical structure of pure histamine (**A**) and the prediction vector plots for histamine in raw (**B**) and processed tuna fish (**C**) regression models.

**Figure 4 foods-10-00885-f004:**
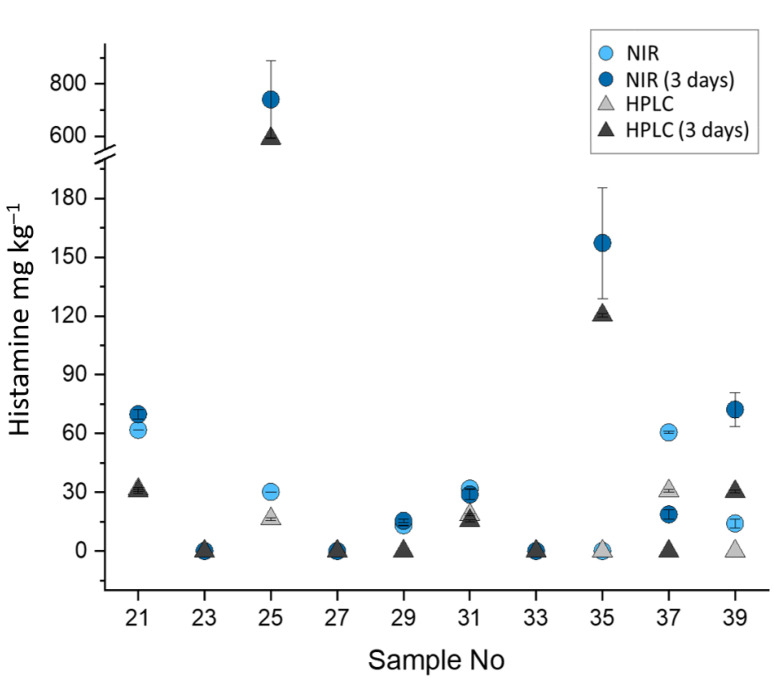
Histamine quantification results in raw tuna fish by NIR and HPLC before and after storage at 20 ± 1 °C for 3 days.

**Table 1 foods-10-00885-t001:** Main quantitative and qualitative (discriminant) applications of NIR spectroscopy to different fish and seafood species.

Dataset	Raw Tuna	Processed Tuna
OPLSR Factors (*p* + *o*)	1 + 4	1 + 5
R^2^X	0.698	0.784
Q^2^	0.915	0.949
RMSECV (mg kg^−1^)	6	4
RMSEE (mg kg^−1^)	6	5
*r* ^2^	0.987	0.989
Slope	1	1
Intercept	1.768 × 10^−5^	−2.942 × 10^−7^
LOD (mg kg^−1^)	4	2
LOQ (mg kg^−1^)	12	8
Repeatability (RSD%)	4.9	3.2
Reproducibility (RSD%)	8.1	4.8
Accuracy (recovery%)		
10 mg kg^−1^	89.4	86.2
50 mg kg^−1^	104.8	98.2
100 mg kg^−1^	102.6	98.4

O OPLSR Factors (*p* + *o*): *p* = number of predictive OPLSR factors; *o* = number of orthogonal OPLSR factors. R^2^X = cumulative sum of squares of the X-matrix explained by all the factors. R^2^ = cumulative sum of square of the Y-matrix explained by all the factors. Q^2^ = cumulative predictive variation in Y-matrix. RMSECV = root mean square error of cross-validation. RMSEE = root mean square error of estimation. RDS%= relative standard deviation.

**Table 2 foods-10-00885-t002:** Comparison of NIR and HPLC performances in predicting histamine content in testing tuna samples.

Raw Tuna Fish				Processed Tuna Fish			
Sample	NIR Prediction	HPLC Prediction	Absolute Error	Sample	NIR Prediction	HPLC Prediction	Absolute Error
1	<LOQ	<LOQ	–	1	75.67 ± 0.01	60.1 ± 0.4	15.6
2	<LOQ	<LOQ	–	2	102.4 ± 7.9	94.3 ± 1.2	8.1
3	1694 ± 300	1279 ± 300	415	3	36.2 ± 0.4	15.6 ± 0.2	20.6
4	18 ± 4	<LOQ	17.7	4	32 ± 2	14.3 ± 0.2	17.4
5	15.7 ± 0.9	<LOQ	15.7	5	15 ± 3	<LOQ	15.4
6	34.7 ± 0.4	16.4 ± 0.4	18.4	6	10 ± 4	<LOQ	10.0
7	<LOQ	<LOQ	–	7	22.1 ± 0.1	<LOQ	22.1
8	23± 2	<LOQ	23.2	8	26.1 ± 1.4	<LOQ	26.1
9	25 ± 2	<LOQ	24.8	9	52.5 ± 1.0	37.2 ± 0.5	15.3
10	12 ± 3	<LOQ	12.1	10	21.8 ± 1.9	<LOQ	21.8
11	17 ± 5	<LOQ	17.4	11	83.3 ± 0.6	72.4 ± 0.9	10.9
12	17 ± 4	<LOQ	16.6	12	32.2 ± 0.8	15.6 ± 0.2	16.7
13	32.1 ± 0.7	10.9 ± 0.4	21.2	13	24.7 ± 1.5	<LOQ	24.74
14	46.4 ± 1.5	18.4 ± 0.6	27.9	14	41.5 ± 0.4	25.7 ± 0.3	15.8
15	62.5 ± 1.9	<LOQ	62.5	15	41 ± 2	24.3 ± 0.3	16.6
16	36 ± 5	<LOQ	35.7	16	17.6 ± 1.1	<LOQ	17.6
17	30.9 ± 1.4	<LOQ	30.9	17	65.0 ± 1.4	40.3 ± 0.5	24.7
18	28 ± 4	20.8 ± 0.5	7.0	18	<LOD	<LOQ	–
19	<LOQ	<LOQ	–	19	<LOD	0.86 ± 0.01	–
20	<LOD	<LOQ	–	20	<LOQ	<LOQ	8.2
21	61.9 ± 0.1	31.5 ± 1.0	30.3	21	<LOD	<LOQ	–
22	33.3 ± 0.3	<LOQ	33.3	22	30.4 ± 0.7	15.7 ± 0.2	14.8
23	<LOQ	<LOQ	–	23	54 ± 3	40.3 ± 0.5	14.1
24	<LOQ	<LOQ	–	24	50 ± 4	22.4 ± 0.3	27.8
25	30.18 ± 0.02	16.4 ± 0.8	13.8	25	46.1 ± 1.9	21.0 ± 0.3	25.0
26	25 ± 4	11.4 ± 1.0	13.9	26	<LOD	<LOQ	–
27	<LOQ	<LOQ	–	27	<LOQ	<LOQ	–
28	<LOD	<LOQ	–	28	12 ± 3	<LOQ	12.2
29	13.1 ± 0.2	<LOQ	13.1	29	<LOQ	<LOQ	–
30	<LOD	<LOQ	–	30	49.9 ± 0.6	30.3 ± 0.4	19.6
31	31.89 ± 0.08	18.5 ± 0.5	13.4	31	56.6 ± 0.1	40.9 ± 0.5	15.7
32	<LOQ	<LOQ	–	32	60.5 ± 0.7	30.5 ± 0.4	30.1
33	<LOQ	<LOQ	–	33	45.8 ± 1.5	25.4 ± 0.3	20.4
34	<LOD	<LOQ	–	34	46.8 ± 1.3	20.7 ± 0.3	26.1
35	<LOQ	<LOQ	–	35	30.6 ± 0.4	15.7 ± 0.2	14.9
36	<LOQ	<LOQ	–	36	37.2 ± 0.8	16.3 ± 0.2	20.9
37	60.6 ± 0.5	30.8 ± 0.6	29.9	37	29.4 ± 0.5	<LOQ	29.4
38	61.2 ± 1.9	27.1 ± 0.5	34.1	38	21.5 ± 0.5	<LOQ	24.5
39	14 ± 2	<LOQ	14.0	39	11.7 ± 1.9	<LOQ	11.7
40	<LOQ	<LOQ	–	40	<LOQ	<LOQ	–

Means and standard deviations results from three and five replicates for HPLC and NIR spectroscopy, respectively. Concentrations are expressed as mg kg^−1^. LOD_(NIR)_ = 3.5 mg kg^−1^ (raw tuna), 2.4 mg kg^−1^ (processed tuna); LOQ_(NIR)_ = 11.6 mg kg^−1^ (raw tuna), 8.6 mg kg^−1^ (processed tuna); LOD_(HPLC)_ = 0.003 mg kg^−1^; LOQ_(HPLC)_ = 0.01 mg kg^−1^.

## Data Availability

The data presented in this study are available on request from the corresponding author.

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
