# Peer review of "Histamine Control in Raw and Processed Tuna: A Rapid Tool Based on NIR Spectroscopy"

_foods, 2021, doi:10.3390/foods10040885_

Round 1
Reviewer 1 Report
Reviewer comments and suggestions
Manuscript ID: foods-1176393
With this paper, the authors study and discuss the issue of biogenic amines in raw and processed tuna fish and the ability of these compounds influence the public health due to their physiological and toxicological effects, being employed as quality indexes.
The objective of this paper is to evaluate if near infrared (NIR) spectroscopy with minimal sample processing could be a suitable technique to rapidly measure the histamine levels in tuna fish. The discussion of this theme presenting easy and quick techniques is very important for food industry because an efficient quality control is needed. The topic of biogenic amines is a very important issue with lots of interested from academia and industry.
In my opinion, the manuscript needs revision, because some information should be completed and presented in a different manner.
General comments on the whole text:
- Please check the punctuation and spaces in the text.
- Check the text to the end of the line does not leave alone digits or values separated from the unit.
- Check the text in terms of language, so as to constantly keep one time.
- The importance of this subject must be strongly supported, showing the importance of monitoring these compounds in order to increase the consumers’ acceptability, because consumers are unable to reject these food products based on sensorial parameters.
Some concrete comments would be as follows:
- The introduction must clearly state the originality of this paper and the impact of this study on the fish industry.
- It would be interesting to discuss the differences and considerations for using different techniques for this quantification.
- The paper is well written, but the authors should emphasize the importance and the advantages of NIR spectroscopy over other techniques. The future trends and perspectives should be discussed.
- The authors should emphasize that the quantity of amines in food depends not only on the amount of microorganisms present, but on the activity of the decarboxylase enzyme on specific amino acids and the favourability of the enzymatic conditions, like as, pH, temperature, …. And once these biogenic amines are produced, they are very difficult to destroy by either pasteurization or cooking methods.
- The materials and methods sections are well-described, but they need to be more improved in some points, like as the sample preparation used for each technique.
- The authors have to explore and discuss better the following results “While further studies are needed to deepen these observations and to assess the impact of the overestimation of the histamine amounts in tuna by NIR spectroscopy,….”. because for an industry this “signal amplification” could be a problem.
- The presentation quality of figure 2 must be improved. The layout of table 2 is confusing, so it will be better to separate the results of raw tuna samples from processed tuna samples in the same table.
- Ensure that all references are the most recent and relevant to the arguments in the paper.
Author Response
GENERAL STATEMENT: With this paper, the authors study and discuss the issue of biogenic amines in raw and processed tuna fish and the ability of these compounds influence the public health due to their physiological and toxicological effects, being employed as quality indexes.
The objective of this paper is to evaluate if near infrared (NIR) spectroscopy with minimal sample processing could be a suitable technique to rapidly measure the histamine levels in tuna fish. The discussion of this theme presenting easy and quick techniques is very important for food industry because an efficient quality control is needed. The topic of biogenic amines is a very important issue with lots of interested from academia and industry.
In my opinion, the manuscript needs revision, because some information should be completed and presented in a different manner.
RESPONSE: The authors thank the reviewer for the thorough revision that helped to increase the quality of the present manuscript, with the hope that the following revisions are satisfactory to meet the reviewer’s issues. All the reviewer's suggestions and comments have been addressed, providing below a point-by-point response. All the revised parts of the manuscript have been reported in red color.
GENERAL COMMENTS ON THE WHOLE TEXT
COMMENT 1: Please check the punctuation and spaces in the text.
RESPONSE 1: Typing errors have been checked and corrected throughout the manuscript.
COMMENT 2: Check the text to the end of the line does not leave alone digits or values separated from the unit.
RESPONSE 2: The manuscript has been checked and numerical values separated from the units or isolated digits have been corrected. In addition, according to 9th edition of the International System of Units (SI) Brochure (https://www.bipm.org/utils/common/pdf/si-brochure/SI-Brochure-9.pdf) no periods are present after the unit symbols (except at the end of a sentence) and a space is always used to separate the unit from the number (except for superscript units for plane angle, °, and percentage unit, %).
COMMENT 3: Check the text in terms of language, so as to constantly keep one time.
RESPONSE 3: The manuscript has been edited by a native English speaker. Grammar and style errors have been corrected.
COMMENT 4: The importance of this subject must be strongly supported, showing the importance of monitoring these compounds in order to increase the consumers’ acceptability, because consumers are unable to reject these food products based on sensorial parameters
RESPONSE 4: The suggestion proposed by the reviewer has been accepted and a focus on the lack of perception of histamine by human senses has been added to the text, by remarking that this aspect could substantially aggravate the risk of exposure of consumers to histamine (page 2, lines 57-60 of the revised manuscript).
SOME CONCRETE COMMENTS WOULD BE AS FOLLOWS:
COMMENT 5: The introduction must clearly state the originality of this paper and the impact of this study on the fish industry.
RESPONSE 5: The significance and the impact of the present research work to the fish industries had been already provided in the Introduction section (page 3, lines 118-122 of the revised manuscript). To emphasize the novelty of the research, the authors have now specified that this is the first time a similar approach has been used for the direct quantification of histamine in tuna fish (page 3, line 116 of the revised manuscript) and that minimal sample preparations has been applied (page 3, line 118 of the revised manuscript), thus leading to sample saving and reduced workload which are particularly advantageous for fish industries and fish markets. For the sake of conciseness and to limit the length of the Introduction section, advantages for the fish industry had been also discussed in a more detailed way in the Conclusion section (page 13, lines 465-469; lines 475-479 of the revised manuscript).
COMMENT 6: It would be interesting to discuss the differences and considerations for using different techniques for this quantification.
RESPONSE 6: The suggestion of the reviewer has been followed and it has been concisely specified that the other analytical techniques commonly employed for the detection and the quantification of histamine are known to share certain common limitations, such as costliness of the overall instrumentation and/or regents, high consumptions of solvents and samples, destructiveness of the samples, longer preliminary sample preparation steps, longer time for analysis, and the need for qualified personnel in the execution of the analyses (page 2, lines 78-85 of the revised manuscript).
COMMENT 7: The paper is well written, but the authors should emphasize the importance and the advantages of NIR spectroscopy over other techniques. The future trends and perspectives should be discussed.
RESPONSE 7: The advantages of NIR spectroscopy over other analytical techniques from the perspectives of rapidity, non-destructiveness, cost-effectiveness, and high throughput had been already reported at the end of the Introduction section (page 3, lines 118-122) and in the Conclusion section (page 13, lines 459-461, lines 465-469 of the revised manuscript). To better emphasize the benefits of using NIR spectroscopy, some concise background information about the drawbacks/limitations of the analytical techniques for histamine quantification (other than NIR spectroscopy) has been now added to the text (page 2, lines 78-85 of the revised manuscript).
The short- and long-term perspectives of the proposed methodology had been reported as concluding remarks in the Conclusion section of the manuscript (page 13, lines 469-481 of the revised manuscript).
COMMENT 8: The authors should emphasize that the quantity of amines in food depends not only on the amount of microorganisms present, but on the activity of the decarboxylase enzyme on specific amino acids and the favorability of the enzymatic conditions, like as, pH, temperature, …. And once these biogenic amines are produced, they are very difficult to destroy by either pasteurization or cooking methods.
RESPONSE 8: As suggested by the reviewer, the factors influencing the formation of histamine in fish muscle have been now discussed in a more detailed way. Specifically, information about free histidine amounts and release of microbial decarboxylase enzymes, as well as the effects exerted by the composition of matrix, microorganisms and handling/storage conditions of the products have been provided (pages 1-2, lines 45-54 of the revised version of the manuscript.
At the same time, the thermostability of histamine and, thus, problems related to its removal from fish matrices has been provided (page 2, lines 54-56 of the revised manuscript).
To support all the above statements, two new and recent references have also been added to the text (new ref. No. 8, Capillas, C.; Herrero, A. M. Impact of biogenic amines on food quality and safety. Foods 2019, 8, 62, doi:10.3390/foods8020062; new ref. No 10, Durak-Dados, A.; Michalski, M.; Osek, J. Histamine and other biogenic amines in food. J. Vet. Res. 2020, 64, 281, doi:10.2478/jvetres-2020-0029).
COMMENT 9: The materials and methods sections are well-described, but they need to be more improved in some points, like as the sample preparation used for each technique.
RESPONSE 9: According to the reviewer ‘s remark, a new paragraph concerning the preparation of tuna samples for HPLC analysis has been now added to the Materials and Methods section (page 4, lines 160-169 of the revised manuscript). Specifically, information about sample amounts, reagents and solvents, as well as derivatization procedure have been provided.
Sample preparation for NIR analysis had been already discussed in the Material and Methods section. In this case, a detailed description of the whole procedure had been reported in Section 2.2 (page 3, lines 141-150) and in Section 2.4 (page 4, lines 177-183 of the revised manuscript). In order to make this information clearer and more immediate, a flowchart summarizing the principal experimental procedure steps had been also created and provided in Figure 1 (page 5 of the revised manuscript).
COMMENT 10: The authors have to explore and discuss better the following results “While further studies are needed to deepen these observations and to assess the impact of the overestimation of the histamine amounts in tuna by NIR spectroscopy,….”. because for an industry this “signal amplification” could be a problem.
RESPONSE 10: In accordance with the kind suggestion of the reviewer, the paragraph at the end of the Discussion section has now been modified to better clarify the author’s point of view (page 13, lines 445-450).
The tendency of NIR spectroscopy to overestimate results related to the quantification of food compounds in small quantities was previously reported in literature and associated to possible interferences with other minor compounds of the food matrix (pages 12-13, lines 436-440 of the revised manuscript). At the same time, the problem of overestimation of the analyte is also typical of other widely used analytical techniques for histamine quantification. This is the case of the enzyme-based methods [e.g., ELISA) (Ben-Gigirey, B., C. Craven, and H. An. 1998. Histamine formation in albacore muscle analyzed by AOAC and enzymatic methods. J. Food Sci. 63:1-5; Velenzano, S.; Lippolis, V.; Pascale, M.; Maragos, C. M.; Suman, M.; Visconti, A.; Determination of deoxynivalenol in wheat bran and whole-wheat bran flour by fluorescence polarization immunoassay. Food Anal. Methods 2013, 7, 806-813, doi:10.1007/s12161-013-9684-7) which represent one of the most frequently used tests at industrial level for the detection of histamine, and which require confirmation of the results with the reference method.
Based on the authors’ opinion and experience, if NIR spectroscopy is used as a screening tool to analyze as many fish samples as possible directly on-site, the advantages might overcome the drawbacks associated to the magnification of the amount of histamine in fish. Specifically, the achieved results highlighted that accuracy in predicting histamine was excellent, since the root mean square errors associated to calibration and the validation of the predictive models (i.e., RMECV, RMEE and RMSEP) were globally lower than 6 mg kg-1 on a scale of quantification from 0 to 1000 mg kg-1. Considering that, whatever the specific values, a maximum of 200 mg kg-1 of histamine is tolerated by EU in fresh fish and that the alert limit for histamine content in fish samples should be around 50 mg kg-1, at that range of concentrations the estimation of histamine by NIR deviated on average 20 mg kg-1 from that measured by HPLC (Table 2, page 11 of the revised manuscript). Therefore, this deviation may be to some extent problematic for industry since it would increase the number of suspicious samples to be confirmed by reference methods, but, at the same time, would be more beneficial than an underestimation due to the improvement and the quality and safety standards. The latter point had been already discussed in the Conclusion section (page 13, lines 462-474 of the revised manuscript).
COMMENT 11: The presentation quality of figure 2 must be improved. The layout of table 2 is confusing, so it will be better to separate the results of raw tuna samples from processed tuna samples in the same table.
RESPONSE 11: Figure 2 has been now modified to increase the overall dimension, the quality and the resolution (page 9 of the revised manuscript).
The presentation of Table 2 has also been modified. In particular, the overall width of the Table 2 has been increased (page 11 of the revised manuscript) so as to put numbers referring to one measurement i.e. (mean values and standard deviations) in a single row and, thus, to improve the overall readability. At the same time and in accordance with the comments of the other reviewer, the significant digits of the uncertainty values have been rounded to one single digits and mean values have been rounded accordingly. This way, numerical digits have been reduced and the data presented in more orderly way.
COMMENT 12: Ensure that all references are the most recent and relevant to the arguments in the paper.
RESPONSE 12: The most recent and pertinent references to the subjects dealt within the present research article have been chosen and reported. Moreover, three new and updated references have now been provided to support the new information added to the manuscript:
- [8]: Capillas, C.; Herrero, A. M. Impact of biogenic amines on food quality and safety. Foods 2019, 8, 62, doi:10.3390/foods8020062).
- Ref [10]: Durak-Dados, A.; Michalski, M.; Osek, J. Histamine and other biogenic amines in food. J. Vet. Res. 2020, 64, 281, doi:10.2478/jvetres-2020-0029
- Ref [18]: Surya, T.; Sivaraman, B.; Alamelu, V.; Priyatharshini, A.; Arisekar, U.; Sundhar, S. Rapid methods for histamine detection in fishery products. Int. J. Curr. Microbiol. Appl. Sci 2019, 8, 2035-2046, doi:10.20546/ijcmas.2019.803.242.
Reviewer 2 Report
the article is well written, compares a standard and a progressive method for biogenic amines determination. The application of NIRS is interesting both for experimental science and for practical (fast) applications.
However, I would recommend improvement of presentation the numbers in tables. Experimental uncertainties should almost always be rounded to one significant figure.
This rule has only one significant exception. If the leading digit in the uncertainty is "1", then keeping two significant figures in the uncertainty may be better.
In Table 2 : e.g.:
Sample 3: 1279+/-338 ..... corr.: 1279+/-300
Sample 6: 34.74+/-0.38 ...... corr.: 34.7+/-0.4
Sample 13: 32.09+/-0.71 ..... corr.: 32.1+/-0.7
Rule exception example:
Sample 14: 46.35+/-1.51 ..... corr.: 46.4+/-1.5
Author Response
GENERAL STATEMENT: the article is well written, compares a standard and a progressive method for biogenic amines determination. The application of NIRS is interesting both for experimental science and for practical (fast) applications.
However, I would recommend improvement of presentation the numbers in tables. Experimental uncertainties should almost always be rounded to one significant figure.
This rule has only one significant exception. If the leading digit in the uncertainty is "1", then keeping two significant figures in the uncertainty may be better.
In Table 2 : e.g.:
Sample 3: 1279+/-338 ..... corr.: 1279+/-300
Sample 6: 34.74+/-0.38 ...... corr.: 34.7+/-0.4
Sample 13: 32.09+/-0.71 ..... corr.: 32.1+/-0.7
Rule exception example:
Sample 14: 46.35+/-1.51 ..... corr.: 46.4+/-1.5
RESPONSE: The authors gratefully acknowledge the reviewer for the positive feedback and the efforts dedicated to the reading and improvement of the present research article. Following the reviewer’s kind suggestion, the presentation of the RMEECV, RMSEE, LOD, and LOQ values (mg kg-1) as well as mean values and the standard deviations reported in Table 2 have been modified (page 11 of the revised manuscript). Only one significant digit was retained when expressing the uncertainties, while two significant digits have been reported when uncertainties values were characterized by the presence of “1” as leading digits of the number. Mean values have been also rounded to the same order of magnitude (same decimal positions) as the corresponding uncertainty values.
Many other parts of the manuscripts have been modified to meet the reviewer’s suggestions and reported in red color throughout the text.
Round 2
Reviewer 1 Report
The authors replied to my comments and they have provided a new and improved version of the paper.